# Provably efficient, succinct, and precise explanations

**Guy Blanc**[*]
Stanford University
gblanc@stanford.edu

**Jane Lange**[*]
Massachusetts Institute of Technology
jlange@mit.edu

**Li-Yang Tan**[*]
Stanford University
liyang@cs.stanford.edu

## Abstract

We consider the problem of explaining the predictions of an arbitrary blackbox model $f$: given query access to $f$ and an instance $x$, output a small set of $x$'s features that in conjunction essentially determines $f(x)$. We design an efficient algorithm with provable guarantees on the succinctness and precision of the explanations that it returns. Prior algorithms were either efficient but lacked such guarantees, or achieved such guarantees but were inefficient.

We obtain our algorithm via a connection to the problem of *implicitly* learning decision trees. The implicit nature of this learning task allows for efficient algorithms even when the complexity of $f$ necessitates an intractably large surrogate decision tree. We solve the implicit learning problem by bringing together techniques from learning theory, local computation algorithms, and complexity theory.

Our approach of "explaining by implicit learning" shares elements of two previously disparate methods for post-hoc explanations, global and local explanations, and we make the case that it enjoys advantages of both.

## 1 Introduction

Modern machine learning systems have access to unprecedented amounts of computational resources and data, enabling them to rapidly train sophisticated models. These models achieve remarkable performance on a wide range of tasks, but their success appears to come at a price: the complexity of these models, responsible for their expressivity and accuracy, makes their inner workings inscrutable to human beings, rendering them powerful but opaque blackboxes. As these blackboxes become central in mission-critical systems and their predictions increasingly relied upon in high-stakes decisions, there is a growing urgency to address their lack of interpretability [DVK17, Lip18].

There has therefore been a surge of interest in the problem of *explaining* the predictions of blackbox models: *why* did a model $f$ assign an instance $x$ the label $f(x)$? While there are numerous possibilities for what qualifies as an explanation (see e.g. [SK10, BSH+10, SVZ14, RSG16, KL17, LL17, STY17]), in this work we consider an explanation to be a set of $x$'s features that in conjunction essentially determines $f(x)$ [RSG18]. Following terminology from complexity theory, we call such an explanation a *certificate*.

We seek *succinct* and *precise* certificates. Intuitively, this means that we would like the set of features to be as small as possible, and for it to nonetheless be a sufficient explanation for $f(x)$ with high probability; more formally, we have the following definition:

---

[*]Alphabetical order.

**Definition 1** (Succinct and precise certificates). *Let $f : \{\pm 1\}^d \to \{\pm 1\}$ be a classifier and $x \in \{\pm 1\}^d$ be an instance. We say that a set $C \subseteq [d]$ of features is a size-$k$ $\varepsilon$-error certificate for $x$ if both of the following hold:*

- *Succinctness: $|C| \leq k$,*

- *Precision: $\Pr_{\boldsymbol{y} \sim \{\pm 1\}^d} \big[ f(\boldsymbol{y}) \neq f(x) \mid \boldsymbol{y}_C = x_C \big] \leq \varepsilon$,*

*where we write '$\boldsymbol{y}_C = x_C$' to mean that $\boldsymbol{y}$ and $x$ agree on all the features in $C$.*

**Our main result.** We give an efficient certificate-finding algorithm with provable guarantees on the succinctness and precision of the certificates that it returns. Our algorithm is model agnostic, requiring no assumptions about the structure of $f$.

**Theorem 1.** *Our algorithm $\mathscr{A}$ is given as input an instance $x$ and parameters $\varepsilon, \delta \in (0, 1)$. It makes queries to a blackbox model $f$ and returns a certificate $\mathscr{A}(x)$ for $x$ with the following guarantees.*

*With probability $1 - \delta$ over a uniform random instance $\boldsymbol{x} \sim \{\pm 1\}^d$, $\mathscr{A}(\boldsymbol{x})$ is an $\varepsilon$-error certificate of size $\mathrm{poly}(\mathcal{C}(f), 1/\varepsilon, 1/\delta)$, where $\mathcal{C}(f)$ is the "average certificate complexity" of $f$. The time and query complexity of $\mathscr{A}$ is $\mathrm{poly}(d, \mathcal{C}(f), 1/\varepsilon, 1/\delta)$.*

There is a sizable literature on certificate finding and related problems, studying them from both algorithmic and hardness perspectives. Previous algorithms, which we overview next, were either efficient but lacked provable guarantees on the succinctness and precision of the certificates that they return, or achieved such guarantees but were inefficient. Furthermore, our algorithm circumvents several hardness results for finding succinct and precise certificates, which we also discuss next.

## 1.1 Prior work on certificate finding

**Efficient heuristics.** Recent work of Ribeiro, Singh, and Guestrin [RSG18] studies certificates (which they term *anchors*) from an empirical perspective. They demonstrate, through experiments and user studies, the effectiveness of certificates as explanations across a variety of domains and tasks. Their work highlights the ease of understanding of certificates by human beings and their clarity of scope. The authors also point out several advantages of certificates over LIME explanations [RSG16].

Their work gives an efficient heuristic, based on greedy search, for finding high-precision certificates. However, there are no guarantees on the succinctness of these certificates, and in fact, it is easy to construct classifiers $f : \{\pm 1\}^d \to \{\pm 1\}$ with near-minimal certificate complexity, $\mathcal{C}(f) = 2$, for which their heuristic returns certificates of near-maximal size, $\Omega(d)$. (We elaborate on this in the body of this paper.) This should be contrasted with the guarantees of Theorem 1; specifically, the dimension-independent bound on the sizes of the certificates that our algorithm returns.

**Prime implicants.** A separate line of work has focused on finding *prime implicants* [Ign20, DH20, INM19, INMS19, SCD18]. In the terminology of Definition 1, an implicant is a 0-error certificate, and an implicant is prime if its error increases whenever a single feature is removed from it. We note that an implicant being prime (i.e. of minimal size) is not equivalent to it being the most succinct (i.e. of minimum size): a classifier $f : \{\pm 1\}^d \to \{\pm 1\}$ can have an implicant of size 1 and also prime implicants of size $d - 1$.

While 0-error certificates are desirable for their perfect precision, it is often impossible to find them efficiently. With no assumptions on $f$, even *verifying* that a certificate has 0 error requires querying $f$ on the potentially exponentially many possible instances consistent with that certificate. Existing algorithms therefore focus on specific model classes. For example, [DH20] gives an algorithm for enumerating all prime implicants for a restricted class of circuits. Still, there are numerous hardness results even for model-specific algorithms. For CNF formulas, determining whether a certificate is 0-error is NP-complete. For general size-$m$ circuits, finding the an 0-error certificate that has size within $m^{1-\varepsilon}$ of the smallest possible is $\mathrm{NP}^{\mathrm{NP}}$-complete for any $\varepsilon > 0$ [Uma01].

Setting aside the computational intractability of finding 0-error certificates, we note that they are in general much less succinct than $\varepsilon$-error certificates. It is easy to construct examples where a size-1 certificate with 0.01-error exists, but the only 0-error certificate is the trivial one of size $d$ containing

all features. In general, there is a natural tradeoff between the two desiderata of succinctness and precision, and our algorithm allows the user to choose their desired tradeoff rather than forcing them to choose perfect precision at the cost of succinctness.

**Hardness of finding approximate certificates.** There are also intractability results for finding the smallest $\varepsilon$-error certificate. [WMHK21] show that for any $\varepsilon$, determining whether there exists an $\varepsilon$-error certificate of size $k$ for a given circuit and instance is $\mathrm{NP}^{\mathrm{PP}}$-complete. Furthermore, they show that assuming $\mathrm{P} \neq \mathrm{NP}$, there is no efficient algorithm that can even approximate the size of the smallest $\varepsilon$-error certificate to within a factor of $d^{1-\alpha}$ for any $\alpha > 0$.

Theorem 1 circumvents these hardness results because our algorithm is only expected to succeed for most (at least a $1 - \delta$ fraction) rather than all instances, and only returns a small certificate relative to the average certificate complexity of the model, rather than smallest for a particular instance.

## 1.2 Our approach and techniques

We connect certificate finding to a new algorithmic problem that we introduce, that of *implicitly* learning decision trees. The key to this connection is a deep result from complexity theory, Smyth's theorem [Smy02], which enables us to relate the certificate and decision tree complexities of functions. We then show how recently developed decision tree learning algorithms [BLT20b, BGLT20] can be extended to solve the implicit learning problem. In more detail, there are three modular components to our approach:

○ *Implicitly learning decision trees.* Decision trees are the canonical example of an interpretable model. Their predictions admit simple explanations: a certificate for an instance $x$ is the root-to-leaf path in the tree that $x$ follows. A natural and well-studied approach to explaining a blackbox model $f$ is therefore to first learn a decision tree $T$ that well-approximates $f$, and with this surrogate decision tree $T$ in hand, one can then output a certificate for any instance $x$ by returning the corresponding path in $T$ [CS95, BS96, VAB07, ZH16, VLJ$^+$17, BKB17, VS20]. A limitation of this approach lies in the fact that many models of interest are inherently complex and cannot be well-approximated by a decision tree of tractable size.

To circumvent this, we introduce the problem of *implicitly* learning decision trees. Roughly speaking, an algorithm for this task allows one to efficiently navigate a surrogate decision tree $T$ for $f$—*without building $T$ in full*. With such an algorithm, the complexity of finding a certificate scales with the *depth* of $T$, making it exponentially more efficient than building $T$ in full, the complexity of which scales with its overall *size*.

○ *Relating certificate and decision tree complexities.* To translate algorithms for implicitly learning decision trees into algorithms for finding certificates, we apply a theorem of Smyth that relates the certificate complexity of a function to its decision tree complexity. The notion of certificates is central to complexity theory, where it is basis of the complexity class NP. (Smyth's result resolved a longstanding conjecture of Tardos [Tar89] that was motivated by the relationship between P and NP $\cap$ coNP.)

○ *An efficient algorithm with provable guarantees.* With the two items above in hand, we are able to leverage recent advances in decision tree learning to design certificate-finding algorithms. Specifically, we show that the decision tree learning algorithm of Blanc, Gupta, Lange, and Tan [BGLT20] can be extended to the setting of implicit learning. Our resulting certificate-finding algorithm is simple: it constructs a certificate $C$ for an instance $x$ by recursively adding to $C$ the most *noise stabilizing* feature. Fruitful connections between noise stability and learnability have long been known [KOS04, KKMS08]; our work further demonstrates its utility for certificate finding.

Our overall approach falls within the framework of *Local Computation Algorithms* [RTVX11]. Such algorithms solve computational problems for which the output—in our case, the surrogate decision tree—is so large that returning it in its entirety would be intractable. Local computation algorithms are able access and return only select parts of the output—in our case, the path through the tree that corresponds to the specific instance $x$ of interest—efficiently and consistently.

## 1.3 Discussion of broader context: global and local explanations

Existing approaches to post-hoc explanations mostly fall into two categories. *Global explanations* seek to capture the behavior of the entire model $f$, often by approximating it with a simple and interpretable model such as a decision tree [CS95, BS96, VAB07, ZH16, VLJ$^+$17, BKB17, VS20] or a set of rules [LKCL19, LAB20]. A limitation of such approaches, alluded to above and also discussed in numerous prior works (see e.g. [RSG16, RSG18]), is that complex models often cannot be well-approximated by simple ones. In other words, using a simple surrogate model necessarily results in low fidelity to the original model.

Our work falls in the second category of *local explanations* [SK10, BSH$^+$10, SVZ14, RSG16, KL17, LL17, RSG18]. These seek to explain $f$'s label for specific instances $x$. Several of these approaches are based on notions of $f$ being "simple around $x$": for example, LIME explanations [RSG16] show that $f$ is "approximately linear around $x$", and certificates, the focus of our work, show that $f$ is "approximately constant in a subspace containing $x$". The corresponding algorithms can therefore be run on models that are too complex to be faithfully represented by a simple global surrogate model.

While our work falls in the category of local explanations, we believe that our new approach of "explaining by implicit learning" enjoys advantages of both local and global methods. The local explanations that our algorithm returns are all consistent with a *single* decision tree $T$ that well-approximates $f$ globally. The implicit nature of the learning task allows for $T$ to be intractably large, hence allowing for corresponding algorithms to be run on complex models $f$, circumventing the limitation of global methods discussed above. On the other hand, the existence of a single global surrogate decision tree, albeit one that may be too large to construct in full, affords several advantages of global methods. We list a few examples:

- *Partial information about global structure.* Our implicit learning algorithm can efficiently construct the subgraph of $T$ comprising the root-to-leaf paths of a few specific instances; alternatively, it can construct the subtree of $T$ rooted at a certain node, or the first few layers of $T$. All of these could shed light on the global behavior of $f$.

- *Feature importance information.* Our implicit decision tree $T$ has useful properties beyond being a good approximator of $f$. As we will show, its structure carries valuable semantic information about $f$, since the feature queried at any internal node $v$ is the most "noise stabilizing" feature of the subfunction $f_v$. The features in the certificates that our algorithm returns can be ordered accordingly, each being the most noise stabilizing feature of the subfunction determined by $x$'s value on the previous features.

- *Measures of similarity between instances.* Every decision tree $T$ naturally induces a "similarity distance" between instances, given by the depth of their lowest common ancestor within $T$. Therefore pairs of instances that share a long common path in $T$ before diverging (or do not diverge at all) are considered very similar, whereas pairs of instances that diverge early on, say at the root, are considered very dissimilar. (Such tree-based distance functions have been influential in the study of hierarchical clustering; see e.g. [Das16].) This distance between two instances can be easily calculated from the certificates that our algorithm returns for them.

## 1.4 Preliminaries

**Feature and distributional assumptions.** We focus on binary features and the uniform distribution over instances. Several aspects of our approach extend to more general feature spaces and distributions; we elaborate on this in the conclusion. We use **boldface** to denote random variables (e.g. $\boldsymbol{x} \sim \{\pm 1\}^d$), and unless otherwise stated, all probabilities and expectations are with respect to the uniform distribution.

**Decision tree and certificate complexity.** The *depth* of a decision tree is the length of the longest root-to-leaf path, and its *size* is the number of leaves.

**Definition 2** (Decision tree complexity)**.** *The $\varepsilon$-error decision tree complexity of $f : \{\pm 1\}^d \to \{\pm 1\}$, denoted $\mathcal{D}(f, \varepsilon)$, is the smallest $k$ for which there exists a depth-$k$ decision tree $T$ satisfying* $\Pr[T(\boldsymbol{x}) \neq f(\boldsymbol{x})] \leq \varepsilon$.

**Definition 3** (Certificate complexity). *For a function $f : \{\pm 1\}^d \to \{\pm 1\}$ and an instance $x \in \{\pm 1\}^d$, the $\varepsilon$-error certificate complexity $f$ at $x$, denoted $\mathcal{C}(f, x, \varepsilon)$, is the size of the smallest $\varepsilon$-error certificate for $x$. That is, $\mathcal{C}(f, x, \varepsilon)$ is the size of the smallest set $C \subseteq [d]$ for which*

$$\Pr_{\boldsymbol{y} \sim \{\pm 1\}^d} \left[ f(\boldsymbol{y}) \neq f(x) \mid \boldsymbol{y}_C = x_C \right] \leq \varepsilon.$$

*The $\varepsilon$-error certificate complexity of $f$ is the quantity*

$$\mathcal{C}(f, \varepsilon) \coloneqq \mathbb{E}_{\boldsymbol{x} \sim \{\pm 1\}^d} [\mathcal{C}(f, \boldsymbol{x}, \varepsilon)].$$

*When $\varepsilon = 0$, we simply write $\mathcal{C}(f, x)$ and $\mathcal{C}(f)$.*

## 2 Implicitly learning decision trees

Our motivation for introducing the problem of *implicitly* learning decision trees is based on a simple but key property of decision trees. For any instance $x$, only a tiny portion of $T$'s overall structure is "relevant" for its operation on $x$: the root-to-leaf path in $T$ that $x$ follows. The depth of a decision tree is, in general, exponentially smaller than its overall size, so this is indeed a tiny portion.

This natural modularity of decision trees is perhaps the most fundamental reason that decision trees are so interpretable, and we design our overall approach of "explaining by implicitly learning" to take advantage of it. For contrast, consider polynomials instead of decision trees: for a polynomial $p$ and an instance $x$, all the monomials of $p$ are "relevant" for $p$'s operation on $x$.

An algorithm for implicitly learning decision trees allows one to efficiently *navigate* a decision tree hypothesis for a target function without constructing the tree in full.

**Definition 4** (Implicitly learning decision trees). *An algorithm for implicitly learning decision trees is given query access to a target function $f : \{\pm 1\}^d \to \{\pm 1\}$ and supports the following basic operations on a decision tree hypothesis $T$ for $f$:*

1. *ISLEAF$(T, \alpha)$ which, given some node $\alpha$, returns whether $\alpha$ is a leaf in $T$.*

2. *QUERY$(T, \alpha)$ which, given a non-leaf node $\alpha$, returns the index $i \in [d]$ corresponding to the feature that $T$ queries at node $\alpha$.*

3. *LEAFVALUE$(T, \alpha)$ which, given a leaf node $\alpha$, returns the output value of that leaf.*

*We assume that $\alpha$ is represented as a restriction of a subset of the features $\{\pm 1\}^d$ corresponding to the features queried along to the root-to-$\alpha$ path in $T$.*

The focus of our paper will be on the connections between implicit learning decision trees and our efficiently finding certificates. As discussed in Section 1.3, we believe that the former problem is of independent interest and will see applications beyond certificates; we return to this point in the conclusion.

### 2.1 The connection between implicitly learning DTs and certificate finding

Since an implicit learning algorithm is not required to fully construct the decision tree hypothesis, this definition allows for efficient algorithms even when the complexity of $f$ necessitates a surrogate decision tree of intractably large size. Building on this, we now show that algorithms for implicitly learning decision trees yield certificate-finding algorithms with efficiency that scales with the *depth* of the decision tree hypothesis $T$ rather than its overall size.

**Lemma 2.1** (Implicitly learning decision trees $\Rightarrow$ certificate finding). *Let $f : \{\pm 1\}^d \to \{\pm 1\}$ and $\varepsilon, \delta \in (0, 1)$. Suppose there is algorithm for implicitly learning $f$ where the decision tree hypothesis $T$ satisfies:*

1. *$T$ is $(\varepsilon\delta/2)$-close to $f$, meaning $\Pr_{\boldsymbol{x} \sim \{\pm 1\}^d}[T(\boldsymbol{x}) \neq f(\boldsymbol{x})] \leq \varepsilon\delta/2$.*

2. *$T$ has depth $k$.*

FINDCERTIFICATE$(f, T, x, \varepsilon, \delta)$:

**Given:** Query access to $f : \{\pm1\}^d \rightarrow \{\pm1\}$, an algorithm for implicitly learning $f$ with $T$ as its decision tree hypothesis, instance $x \in \{\pm1\}^d$, precision parameter $\varepsilon$, and confidence parameter $\delta$.

**Output:** An $\varepsilon$-error certificate for $x$ of size at most the depth of $T$, or $\bot$ if no certificate is found.

1. Initialize $\alpha \leftarrow \varnothing$.
2. Initialize $C \leftarrow \varnothing$.
3. While not ISLEAF$(T, \alpha)$:
   (a) Set $i \leftarrow$ QUERY$(T, \alpha)$.
   (b) Add $i$ to $C$.
   (c) Set $\alpha \leftarrow \alpha \cup \{i = x_i\}$.
4. Using queries to $f$, check whether the following holds with confidence at least $1 - \delta$, indicating if $C$ is an $\varepsilon$-error certificate for $x$:

$$\Pr_{\boldsymbol{y} \sim \{\pm1\}^d} \left[ f(\boldsymbol{y}) \neq f(x) \mid \boldsymbol{y}_C = x_C \right] \leq \varepsilon.$$

If so, output $C$. Otherwise, output $\bot$.

Figure 1: How an algorithm for implicitly learning decision trees can be used to design a certificate-finding algorithm.

*Suppose each of the operations in Definition 4 is supported in time $t$. Then there is an algorithm, FINDCERTIFICATE, which on $\boldsymbol{x} \sim \{\pm1\}^d$ runs in time $O(tk) + O(\log(1/\delta)/\varepsilon^2)$ and finds an $\varepsilon$-error certificate of size at most $k$ with probability at least $1 - \delta$.*

*Proof.* Consider the algorithm FINDCERTIFICATE given in Figure 1. By design, the certificates that it returns is $\varepsilon$-error and has size at most $k$, the depth of $T$. Each iteration of the algorithm takes time $O(t)$, and the number of iterations is at most the depth of the root-to-leaf path in $T$ that $x$ follows, which is at most $k$; the additional time complexity of the random sampling step is $O(\log(1/\delta)/\varepsilon^2)$.

It remains to prove that FINDCERTIFICATE$(f, T, \boldsymbol{x}, \varepsilon, \delta)$ outputs $\bot$ with probability at most $\delta$. Let $\mathscr{A}(\boldsymbol{x})$ the certificate checked in Step 4 or FINDCERTIFICATE. We prove that $\mathscr{A}(\boldsymbol{x})$ is an $\varepsilon$-error certificate for $\boldsymbol{x}$ with probability at least $1 - \delta$. First, we union bound the definition of an $\varepsilon$-error certificate as follows:

$$\Pr_{\boldsymbol{y} \sim \{\pm1\}^d} \left[ f(\boldsymbol{y}) \neq f(\boldsymbol{x}) \mid \boldsymbol{y}_{\mathscr{A}(\boldsymbol{x})} = x_{\mathscr{A}(\boldsymbol{x})} \right] \leq \Pr_{\boldsymbol{y} \sim \{\pm1\}^d} \left[ f(\boldsymbol{y}) \neq T(\boldsymbol{y}) \mid \boldsymbol{y}_{\mathscr{A}(\boldsymbol{x})} = x_{\mathscr{A}(\boldsymbol{x})} \right]$$
$$+ \Pr_{\boldsymbol{y} \sim \{\pm1\}^d} \left[ T(\boldsymbol{y}) \neq T(\boldsymbol{x}) \mid \boldsymbol{y}_{\mathscr{A}(\boldsymbol{x})} = x_{\mathscr{A}(\boldsymbol{x})} \right]$$
$$+ \Pr_{\boldsymbol{y} \sim \{\pm1\}^d} \left[ T(\boldsymbol{x}) \neq f(\boldsymbol{x}) \mid \boldsymbol{y}_{\mathscr{A}(\boldsymbol{x})} = x_{\mathscr{A}(\boldsymbol{x})} \right].$$

Our goal is to prove for a random $\boldsymbol{x} \sim \{\pm1\}^d$, the probability that the sum of three terms is more than $\varepsilon$ is at most $\delta$. The second term is simplest: whenever $\boldsymbol{y}_{\mathscr{A}(\boldsymbol{x})} = x_{\mathscr{A}(\boldsymbol{x})}$, $\boldsymbol{y}$ and $\boldsymbol{x}$ visit the same leaf in $T$, so $T(\boldsymbol{y}) \neq T(\boldsymbol{x})$ with probability 0. Hence, if the sum of the three terms is more than $\varepsilon$, it must be the case that either the first term or the third term is more than $\varepsilon/2$. The third term is just $\Pr[T(\boldsymbol{x}) \neq f(\boldsymbol{x})]$, independent of the choice of $\boldsymbol{y}$. This is at most $\frac{1}{2}\varepsilon\delta \leq \frac{1}{2}\varepsilon$ since $T$ is $\frac{1}{2}\varepsilon\delta$ close to $f$. Finally, since

$$\mathbb{E}_{\boldsymbol{x} \sim \{\pm1\}^d} \left[ \Pr_{\boldsymbol{y} \sim \{\pm1\}^d} \left[ f(\boldsymbol{y}) \neq T(\boldsymbol{y}) \mid \boldsymbol{y}_{\mathscr{A}(\boldsymbol{x})} = \boldsymbol{x}_{\mathscr{A}(\boldsymbol{x})} \right] \right] = \Pr_{\boldsymbol{y} \sim \{\pm1\}^d} \left[ f(\boldsymbol{y}) \neq T(\boldsymbol{y}) \right] \leq \frac{1}{2}\delta\varepsilon,$$

the first probability is more than $\varepsilon/2$ with probability at most $\delta$ by Markov's inequality. $\square$

## 3   Certificate complexity, decision tree complexity, and Smyth's theorem

Our notion of certificates as defined in Definition 1 is *local*, specific to each instance $x$, whereas Smyth's theorem concerns a *global* notion of certificates, defined for the entire function $f$.

**Notation.**   Fix a function $f : \{\pm 1\}^d \to \{\pm 1\}$ and let $\mathscr{C}$ be a collection of subsets $C \subseteq [d]$. We write '$x \models \mathscr{C}$' if there exists a 0-error certificate $C \in \mathscr{C}$ for $x$, and we write '$x \not\models \mathscr{C}$' otherwise.

Given two collections $\mathscr{C}_1$ and $\mathscr{C}_{-1}$ of subsets, we define a corresponding function $\Phi_{\mathscr{C}_1, \mathscr{C}_{-1}} : \{\pm 1\}^d \to \{\pm 1, \bot\}$ as follows:

$$\Phi_{\mathcal{C}_1, \mathcal{C}_{-1}}(x) = \begin{cases} 1 & \text{if } x \models \mathscr{C}_1 \text{ and } x \not\models \mathscr{C}_{-1} \\ -1 & \text{if } x \models \mathscr{C}_{-1} \text{ and } x \not\models \mathscr{C}_1 \\ \bot & \text{otherwise.} \end{cases}$$

**Definition 5** (Global certificate complexity). *The global $\varepsilon$-error certificate complexity of $f$ : $\{\pm 1\}^d \to \{\pm 1\}$, denoted $\mathcal{GC}(f, \varepsilon)$, is the smallest $k$ for which there exists two collections $\mathscr{C}_1$ and $\mathscr{C}_{-1}$ of size-$k$ subsets satisfying $\Pr[f(\boldsymbol{x}) \neq \Phi_{\mathscr{C}_1, \mathscr{C}_{-1}}(\boldsymbol{x})] \leq \varepsilon$.*

Recalling Definition 2, it is straightforward to verify that $\mathcal{GC}(f, \varepsilon) \leq \mathcal{D}(f, \varepsilon)$ for all $f : \{\pm 1\}^d \to \{\pm 1\}$ and $\varepsilon > 0$. Briefly, give a depth-$k$ $\varepsilon$-error decision tree $T$ for $f$, takes $\mathscr{C}_1$ to be the collection of size-$k$ certificates corresponding to paths leading to 1-leafs, and $\mathscr{C}_{-1}$ to be the collection of size-$k$ certificates corresponding to paths leading to $-1$-leafs. It is easy to see that $\Pr[f(\boldsymbol{x}) \neq \Phi_{\mathscr{C}_1, \mathscr{C}_{-1}}(\boldsymbol{x})] \leq \varepsilon$.

Smyth [Smy02], resolving a longstanding conjecture of Tardos [Tar89], proved a surprising *converse* to the elementary inequality above:

**Theorem 2.** *For all $f : \{\pm 1\}^d \to \{\pm 1\}$ and $\varepsilon > 0$, we have $\mathcal{D}(f, \varepsilon) \leq O(\mathcal{GC}(f, \varepsilon^3/30)^2/\varepsilon^3)$.*

We derive as a corollary of Smyth's theorem a relationship between certificate and decision tree complexities:

**Corollary 3.1** (Bounding decision tree complexity in terms of certificate complexity). *For all $f : \{\pm 1\}^d \to \{\pm 1\}$ and $\varepsilon > 0$, we have $\mathcal{D}(f, \varepsilon) \leq O(\mathcal{C}(f)^2/\varepsilon^9)$.*

*Proof.* Let $k \coloneqq \mathcal{C}(f)$. By Markov's inequality, we have that $\Pr_{\boldsymbol{x} \sim \{\pm 1\}^d}[\mathcal{C}(f, \boldsymbol{x}) \leq 30k/\varepsilon^3] \geq 1 - (\varepsilon^3/30)$. For every $x \in f^{-1}(1)$ (resp. $x \in f^{-1}(-1)$) that contributes to this probability, we include in $\mathscr{C}_1$ (resp. $\mathscr{C}_{-1}$) its certificate of size $30k/\varepsilon^3$. It follows that $\Pr_{\boldsymbol{x} \sim \{\pm 1\}^d}[f(\boldsymbol{x}) \neq \Phi_{\mathscr{C}_1, \mathscr{C}_{-1}}(\boldsymbol{x})] \leq \varepsilon^3/30$ and hence $\mathcal{GC}(f, \varepsilon^3/30) \leq 30k/\varepsilon^3$. By Smyth's theorem, $\mathcal{D}(f, \varepsilon) \leq O(\mathcal{GC}(f, \varepsilon^3/30)^2/\varepsilon^3) = O(k^2/\varepsilon^9)$ and this completes the proof. $\square$

## 4   An efficient algorithm with provable guarantees

The notion of *noise sensitivity* underlies our implicit learning algorithm and its provable guarantees:

**Definition 6** (Noise sensitivity). *For $f : \{\pm 1\}^d \to \{\pm 1\}$ and $p \in (0, 1)$, the noise sensitivity of $f$ at noise rate $p$ is the quantity*

$$\mathrm{NS}_p(f) \coloneqq \mathop{\mathbb{E}}_{\boldsymbol{x} \sim \{\pm 1\}^n} \left[ \Pr_{\boldsymbol{x}' \sim_p \boldsymbol{x}}[f(\boldsymbol{x}) \neq f(\boldsymbol{x}')] \right],$$

*where $\boldsymbol{x}' \sim_p \boldsymbol{x}$ means that $\boldsymbol{x}'$ is drawn by independently rerandomizing each coordinate of $\boldsymbol{x}$ with probability $p$.*

Strong connections between noise sensitivity and learnability have long been known [KOS04, BOW08, KKMS08, DHK+10, Kan14]. Most relevant for us is the work of Blanc, Gupta, Lange, and Tan [BGLT20], which gives a new decision tree learning algorithm with a noise-sensitivity-based splitting criterion, and shows that it achieves strong provable performance guarantees. Our algorithm is an implicit version of theirs that enjoys the exponential efficiency gains made possible by the implicit setting.

## 4.1 Greedy noise stabilizing decision trees

**Definition 7** (Noise stabilizing score). *For $f : \{\pm 1\}^d \to \{\pm 1\}$ and $p \in (0, 1)$, the noise stabilizing score, or simply the score, of the $i$-th feature with respect to $f$ and $p$ is the quantity:*

$$\text{Score}_f(i, p) := \text{NS}_p(f) - \mathop{\mathbb{E}}_{\boldsymbol{b} \sim \{\pm 1\}}[\text{NS}_p(f_{i=\boldsymbol{b}})],$$

*where $f_{i=b}$ is the restriction of $f$ obtained by fixing the $i$-th feature to $b$.*

We associate with every function $f : \{\pm 1\}^d \to \{\pm 1\}$ its *greedy noise stabilizing decision tree*.

**Definition 8** (Greedy noise stabilizing decision tree). *The greedy noise stabilizing decision tree for $f : \{\pm 1\}^d \to \{\pm 1\}$ at noise rate $p$, denoted $\Upsilon_{f,p}$, is the complete depth-$d$ decision tree where at every internal node $\alpha$, the feature $i$ that is queried is the one with the highest noise stabilizing score with respect to the subfunction $f_\alpha$, the restriction of $f$ by the root-to-$\alpha$ path. Every leaf $\ell$ of $\Upsilon_{f,p}$ is labeled according to $f$'s value for the unique instance that follows the root-to-$\ell$ path.*

$\Upsilon_{f,p}$ has maximum depth $d$, the dimension of $f$'s feature space. Since the succinctness of the certificates that of our certificate-finding algorithm returns scales with the depth of the decision tree hypothesis, we will truncate $\Upsilon_f$ at a much smaller depth $k \ll d$. Furthermore, with query access to $f$ one can only obtain high-accuracy estimates of the noise stabilizing scores of each of its features, and not the exact values of these scores. These algorithmic considerations motivate the following variant of Definition 8:

**Definition 9** (Depth-$k$ $\eta$-approximate greedy noise stabilizing decision tree). *Let $f : \{\pm 1\}^d \to \{\pm 1\}$ and $p \in (0, 1)$. For $k \leq d$ and $\eta \in (0, 1)$, a depth-$k$ $\eta$-approximate greedy noise stabilizing decision tree for $f$ at noise rate $p$, denoted $\Upsilon_{f,p}^{k,\eta}$, is a complete depth-$k$ decision tree where:*

- *At every internal node $\alpha$, the feature $i$ that is queried has noise stabilizing score with respect to $f$ that is within $\eta$ of the highest:*

$$\text{Score}_i(f_\alpha, p) \geq \text{Score}_j(f_\alpha, p) - \eta \quad \text{for all } j \neq i.$$

- *Every leaf $\ell$ is labeled $\text{sign}(\mathbb{E}[f_\ell])$.*

## 4.2 Proof of Theorem 1

We now show that, owing to the simple top-down inductive definition of $\Upsilon_{f,p}^{k,\eta}$, there is an efficient algorithm for implicitly learning any target function $f$ with $\Upsilon_{f,p}^{k,\eta}$ as the decision tree hypothesis.

**Lemma 4.1** (Implicit learning with $\Upsilon_{f,p}^{k,\eta}$ as the decision tree hypothesis). *Let $f : \{\pm 1\}^d \to \{\pm 1\}$ be a function. For $k \leq d$ and $\eta, p \in (0, 1)$, given query access to $f$, all the operations of Definition 4 can be supported in time $O(d^2/\eta^2)$ with $\Upsilon_f^{k,\eta}$ as the decision tree hypothesis.*

*Proof.* Since $\Upsilon_f^{k,\eta}$ is a complete tree of depth $k$, we return TRUE for IsLEAF($\Upsilon_{f,p}^{k,\eta}, \alpha$) iff $\alpha$ corresponds to a path of length exactly $k$. Note that query access to $f$ gives us query access to all its subfunctions $f_\alpha$ for any $\alpha$. Therefore, by standard random sampling arguments, we can an estimate of the noise sensitivity of $f_\alpha$ that is accurate to within $\pm \eta$ w.h.p. using $O(1/\eta^2)$ queries to $f$. This allows us to obtain high-accuracy estimates of the noise stabilizing scores of all the features of $f_\alpha$ in time $O(d^2/\eta^2)$, and hence support QUERY($\Upsilon_{f,p}^{k,\eta}, \alpha$) queries by returning the feature with the highest empirical score. Similarly, we can support LEAFVALUE($\Upsilon_{f,p}^{k,\eta}, \alpha$) queries by approximating $\mathbb{E}[f_\alpha]$ to high accuracy and returning its sign. $\square$

We will need a structural theorem of [BLT20a] that bounds the distance between $f$ and the greedy noise stabilizing decision tree $\Upsilon_{f,p}^{k,\eta}$:

**Theorem 3** (Lemma 2.1 of [BLT20a]). *Let function $f : \{\pm 1\}^d \to \{\pm 1\}$ be a function. Consider $\Upsilon_{f,p}^{k,\eta}$ where*

$$k = O((\mathcal{D}(f, \varepsilon)/\varepsilon)^3), \quad \eta = \Theta(1/k), \quad p = O(\varepsilon/\mathcal{D}(f, \varepsilon)).$$

*Then $\Pr[\Upsilon_{f,p}^{k,\eta}(\boldsymbol{x}) \neq f(\boldsymbol{x})] \leq O(\varepsilon)$.*

We are now ready to put all the pieces together and establish Theorem 1:

*Proof of Theorem 1.* By our corollary to Smyth's theorem, Corollary 3.1, we have the bound $\mathcal{D}(f, \varepsilon\delta) \leq O(\mathcal{C}(f)^2/(\varepsilon\delta)^9)$. Combining this with Theorem 3, we get that

1. $\Pr[\Upsilon_f^{k,\eta}(\boldsymbol{x}) \neq f(\boldsymbol{x})] \leq O(\varepsilon\delta)$, where

2. $k \leq O((\mathcal{D}(f, \varepsilon\delta)/\varepsilon)^3) \leq \text{poly}(\mathcal{C}(f), 1/\varepsilon, 1/\delta)$.

These correspond exactly to the two items in the assumption of Lemma 2.1 with '$T$' being $\Upsilon_{f,p}^{k,\eta}$, and the theorem follows. □

**Comparison with the certificate-finding algorithm of [RSG18].**   We conclude this section by comparing our resulting certificate-finding algorithm to the heuristic proposed in [RSG18].

*Different greedy choice.* Both our algorithm and [RSG18]'s heuristic are greedy in nature. Our algorithm builds a certificate by iteratively adding to it the most noise stabilizing feature of $f$, and recursing on the subfunction obtained by restricting $f$ according to $x$'s value for this feature. Our approach can therefore be viewed as using noise stability as a proxy for progress towards a high-precision certificate. In contrast, [RSG18] takes a more direct approach and iteratively adds to the certificate the feature that results in the largest gain in estimated precision.

*Provable performance guarantees.* [RSG18]'s heuristic is efficient and returns high-accuracy certificates, but it is easy to construct examples showing that it fails to return succinct certificates. As a simple example, consider $f(x) = x_i \oplus x_j$, the parity of two unknown features $i, j \in [d]$. Every instance has a certificate of size two, $C = \{i, j\}$, but since any certificate comprising a single feature $\{k\}$ has the same precision (regardless of whether $k \in \{i, j\}$), [RSG18]'s heuristic may include in its certificate all $d - 2$ irrelevant features. In contrast, since $\mathcal{C}(f) = 2$, Theorem 1 shows that our algorithm returns a high-accuracy certificate of constant-size with high probability. ([RSG18] also considers an extension of their algorithm that incorporates beam search; similar hard functions can be constructed for this extension.)

# 5   Conclusion

Certificates are simple and intuitive explanations that have been shown to be effective across domains and applications [RSG18]. In this work we have designed an efficient certificate-finding algorithm and proved that it returns succinct and precise certificates. Prior algorithms were either efficient but lacked such performance guarantees, or achieved such guarantees but were inefficient. Our algorithm also circumvents known intractability results for finding succinct and precise certificates.

**Limitations of our work.**   The main limitation, and perhaps the most immediate avenue for future work, is the feature and distributional assumptions of Theorem 1. We do not believe that these are inherently necessary for the provable guarantees that we achieve. The main bottleneck to relaxing these assumptions is the decision tree learning algorithm of [BGLT20]: their analysis relies on the assumptions of binary features and the uniform distribution, and consequently, so does our extension of their algorithm to the implicit setting.

Other aspects of our approach go through for more general feature spaces and distributions. Smyth's theorem holds for arbitrary product spaces, and so does our corollary relating decision tree and certificate complexities. The overarching connection between implicitly learning decision trees and certificate finding holds for arbitrary feature spaces and distributions.

**Future directions.**   There are numerous other avenues for future work; we list a few concrete ones:

○ *Improved algorithms and guarantees for specific models.* Our algorithm is model agnostic, requiring no assumptions about the model $f$ that it seeks to explain, and therefore can be run on any model. It would be interesting to develop improved algorithms, or to improve upon the guarantees of our algorithms, for specific classes of models, such as deep neural networks and random forests, by leveraging knowledge of their structure.

- *Instantiating our approach with other notions of feature importance.* Our certificate-finding algorithm proceeds by iteratively adding to the certificate the most noise-stabilizing feature. It would be interesting to analyze natural variants of our algorithm that are based on other notions of feature importance (e.g. Shapley values [LL17]). Can we determine the optimal notion of feature importance that will lead to the most efficient algorithm with the strongest guarantees on the succinctness and precision of the certificates that it returns?

- *Implicit learning of other interpretable models.* As discussed in the introduction, we believe that our overall approach of "explaining by implicit learning" enjoys advantages of both local and global approaches to post-hoc explanations. Can our techniques be extended to give implicit learning algorithms for *generalized* decision trees, ones that branch on predicates more expressive than singleton variables? Can we develop implicit learning algorithms for other interpretable models beyond decision trees?

- *Beyond explainability: implicit decision trees as a robust model.* The motivating application of our work is that of explaining blackbox models, and therefore the key feature of decision trees that we have focused on is their interpretability. Decision trees have advantages beyond interpretability, and it would be interesting to explore further applications of algorithms for implicitly learning decision trees. For example, can our techniques be combined with those of Moshkovitz, Yang, and Chaudhuri [MYC21] to robustify arbitrary models, making them more resilient to noise and adversarial examples?

More broadly, our work is built on new connections between post-hoc explainability and the areas of learning theory, local computation algorithms, and complexity theory. It would be interesting to identify other avenues through which ideas from theoretical computer science can be utilized to contribute to a theory of explainable ML.

## Acknowledgments and Disclosure of Funding

We are grateful to the anonymous reviewers, whose comments and suggestions have helped improve this paper. Guy and Li-Yang are supported by NSF CAREER Award 1942123. Jane is supported by NSF Award CCF-2006664.

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
