# OpenReview forum: "Provably efficient, succinct, and precise explanations"
_NeurIPS.cc/2021/Conference — NeurIPS 2021 Poster_

### Official Review · Reviewer_GLcW · 2021-07-07

**Rating:** 6
**Confidence:** 4

**Summary:**

The paper considers a new algorithm for creating  explanations of predictions of arbitrary black box models.
In this paper, an explanation is a subset of the input features that explain the prediction in the sense that changing the input features,
not in the explanation subset at random, has only a small probability of changing the model prediction.

The paper considers only binary features and binary classification and assumes uniform distribution over the input.
The main result is an algorithm that with probability 1-delta over the random choice of input provides an explanation (subset of features) that is polynomial in the average size of
the smallest explanation when averaged over all inputs (and 1/delta, 1/epsilon where epsilon is error allowed).

The main algorithm assumes an existing decision tree algorithms, and simply builds the path from the root to the leaf where the input data point would end up,
the explanation being the features queried in the nodes on the way. This is possible due to the assumption of uniform distribution over the binary features.

The main component of the theorem is based on a result from complexity theory, that bound certificate complexity (an explanation is a certificate)
of a function, polynomially in terms of the decision tree complexity (depth) of the function, which leads to the polynomial in the bound given.

The second important part is the decision tree construction algorithm required for the explanation algorithm. This methods is based on recent results on decision trees and that
works for only uniform distribution over binary features and binary classes, providing the limiting features of the result shown.
The algorithm is sped up with random sampling which is straightforward by the same assumptions.



**Limitations And Societal Impact:**

Yes

**Main Review:**

Overall the method and framework proposed is well formulated and explained.
In terms of the results there are several limitations that need to be overcome for this to be actually useful for explaining predictions as the paper also mentions.

While creating explanations as considered in this paper for several formulations is computationally hard this is circumvented here which is of course positive.
Another plus is that the paper is well written and the problem well formulated and defined and and easy to follow.

However, the proposed method works only for binary features with uniform distribution over the features and is based on a randomly sampled input to explain.
This makes the main theorem somewhat vague.
The polynomial increase in explanation size may seem theoretically succinct but it is not clear that it is since the polynomial's powers seem non-trivial.
Hence the succinctness of these explanations remain to be seen.
The limits of the results shown in this paper means that they seem of theoretical interest only for now.

**Update**.
I have read all the the reviews and author comments.
Personally, while i think experiments could be interesting i do not mind them not being in the paper. Experiments like this are not easy and including one or two plots may not be enough anyways. Hence i agree with the authors that experiments is a logical next step from here and i do not think the paper should be rejected due to lack of experiments.

I think the approach makes sense and may work well in practice perhaps with some fine-tuning as argued in the rebuttal by the authors. Furthermore the results shown opens the door for several follow up directions the two main directions being actually doing experiments with this approach and improving the conditions for which it may work (as the authors state most of the approach actually generalize to a more general setting).

Hence as it stands now after the rebuttal i will increase my score to a 6.



**Time Spent Reviewing:**

6

---

> ### Author Response · Authors · 2021-08-10
> **Response to Reviewer GLcW**
>
> >Overall the method and framework proposed is well formulated and explained. In terms of the results there are several limitations that need to be overcome for this to be actually useful for explaining predictions as the paper also mentions.
> While creating explanations as considered in this paper for several formulations is computationally hard this is circumvented here which is of course positive. Another plus is that the paper is well written and the problem well formulated and defined and and easy to follow.
>
> Thank you for these kind words!
>
> >The proposed method works only for binary features with uniform distribution over the features and is based on a randomly sampled input to explain.
>
> Indeed our provable guarantees require feature and distributional assumptions; however, a core aspect of our framework is completely general: the reduction from certificate-finding to implicitly learning decision trees (all of section 2; in particular Lemma 2.1). We also note that there is a distribution-free algorithm for (non-implicit) learning of decision trees that runs in quasipolynomial time (Ehrenfeucht and Haussler, Learning decision trees from random examples, Information and Computing 89); however, it is not known to support the implicit-learning operations as efficiently as we require in this work. Finding implicit-learning guarantees for distribution-free algorithms, based on [EH89] or otherwise, is an exciting direction for followup work.
>
>
> >The polynomial increase in explanation size may seem theoretically succinct but it is not clear that it is since the polynomial's powers seem non-trivial.
>
> This is a good point.  Although the focus of this work is theoretical -- our goal is to give a polynomial-time algorithm that finds provably succinct and precise certificates -- we believe there is ample followup experimental work to do. We have given worst-case guarantees in this work, but it would be interesting to see whether in practice the certificates found by our implicit-decision-tree framework are in fact even more succinct than our theoretical bounds.
>
> Furthermore, we remark that our implicit decision tree learning algorithm closely resembles widely-employed decision tree algorithms (ID3, C4.5, CART) which are successful in practice even without strong theoretical guarantees. Our framework can also be instantiated with these algorithms themselves as the implicit decision tree algorithm, which guarantees that the certificates are as succinct as the corresponding paths in the decision tree hypothesis.

---

### Official Review · Reviewer_XbsB · 2021-07-15

**Rating:** 5
**Confidence:** 4

**Summary:**

The authors propose a local explanation method based on certificates, i.e. sets of important features that closely (in conjunction) determine the prediction of a given black-box model. The obtained certificates provide guarantees regarding two sensible properties: succinctness and precision (i.e., intuitively, they remain small/sparse and, with high probability, also locally faithful w.r.t. the black-box model). The authors propose an implicitly learned Decision Tree to obtain certificates. This implicit approach reduces the computation time compared to related methods. The authors further demonstrate the relationship between the complexities of such implicit Decision Trees and the corresponding certificates.

**Ethical Concerns:**

There are no apparent ethical concerns regarding this work.

**Limitations And Societal Impact:**

A limitation of the proposed work is the restriction regarding the possible data distribution, which is addressed by the authors. Another, and in my opinion more severe limitation, is the lack of empirical evaluations. It is unclear how the proposed method behaves in practice and compared to other feature-based explanation techniques.

**Main Review:**

The proposed method is relevant to researchers and practitioners concerned with model-agnostic (local) explainability. Indeed, the discussed certificates are an interesting alternative to popular local attribution methods like SHAP or LIME.
The paper is written well, with only a few minor errors (word repetitions etc.), which could be removed in the final version.
The formalization and theoretical motivation appear to be sound. The noise-sensitivity based splitting criterion is adapted from previous work. While the proposed implicit decision tree learning approach is novel, it raises the question why an explanatory decision tree needs to be learnt implicitly. Efficiency cannot be the only criterion (because similar explanatory mechanisms can also be learnt in an offline setting; their application to locally explain a given instance would still be very efficient).
The proposed method and formalizations assume a Bernoulli target and binary-valued features. Although the authors address this limitation and argue that the approach may be rephrased more generally, this remains to be shown.
The paper is missing an experimental evaluation of the proposed method on real-world data. In particular, the paper would benefit from a comparison between certificates and local attribution methods (e.g. SHAP, LIME), or a comparison with the ‘anchors’ and ‘prime implicants’ methods mentioned in the related works section. I would also like to see how the precision error $\epsilon$ and the succinctness parameter $k$ behave in practice (can I actually set a small $\epsilon$ and still obtain small $k$, i.e. sparse certificates, or vice versa?).
Finally, note that the proposed implicit algorithm will always return certificates of size $k$ or no certificate at all. This is because the implicit tree learning algorithm stops only after having reached a path length of size $k$. Only then do the authors check whether the obtained certificate adheres to the $\epsilon$-precision property (see FindCertificate() algorithm, Figure 1). That is, the proposed algorithm potentially misses smaller certificates along the path to the leaf node, which might already satisfy the precision property. The authors should argue why this is a desired behavior or adjust the algorithm accordingly (e.g. by making algorithm step 4 part of the training loop, i.e. changing it to algorithm step 3(d), in Figure 1).


**Time Spent Reviewing:**

4

---

> ### Author Response · Authors · 2021-08-10
> **Response to Reviewer XbsB**
>
> >While the proposed implicit decision tree learning approach is novel, it raises the question why an explanatory decision tree needs to be learnt implicitly. Efficiency cannot be the only criterion (because similar explanatory mechanisms can also be learnt in an offline setting; their application to locally explain a given instance would still be very efficient).
>
> The aim of a certificate-finding algorithm is to provide some explanatory object for a specific instance $x$ when a full decision tree hypothesis would be impractically large. Certificates scale with the *depth* of the decision tree, rather than its size, and the size of a decision tree can be exponentially larger than its depth. So, part of the assumption of our setting is that some resources are limited (running time, memory, or queries to the model to be explained) and it is advantageous to avoid learning a decision tree hypothesis when a single path is enough to explain an input instance.
>
> >The proposed method and formalizations assume a Bernoulli target and binary-valued features. Although the authors address this limitation and argue that the approach may be rephrased more generally, this remains to be shown.
>
> Indeed our provable guarantees require feature and distributional assumptions; however, a core aspect of our framework is completely general: the reduction from certificate-finding to implicitly learning decision trees (all of section 2; in particular Lemma 2.1). Any learning algorithm that can support the implicit learning operations in an efficient manner can be modified to find certificates efficiently as well. Several widely-used decision tree algorithms, such as ID3, C4.5, and CART, exhibit strong performance in experiments on data with real-valued features and targets and are suitable for implicit learning due to their top-down nature.
>
> While ID3, C4.5, and CART work well on real-world data, they lack provable guarantees. This work focuses on the “noise-stabilizing” implicit learning algorithm because its theoretical guarantees are universal for all functions with a Bernoulli target and binary features, but we expect that the classic top-down algorithms combined with our framework would perform well in experiments.
>
>
> >The paper is missing an experimental evaluation of the proposed method on real-world data.
>
> The focus of our work is to design a polynomial-time algorithm whose certificates are provably succinct and precise. We agree that an in-depth experimental comparison of our algorithm to alternatives is a concrete direction for followup work.
>
> >Finally, note that the proposed implicit algorithm will always return certificates of size $k$ or no certificate at all...the authors should argue why this is a desired behavior or adjust the algorithm accordingly (e.g. by making algorithm step 4 part of the training loop, i.e. changing it to algorithm step 3(d), in Figure 1).
>
> Good point, the algorithm can be modified to terminate once an $\epsilon$-precise certificate is found, using exactly the change you indicated. We will add a remark to this effect in the next revision.

---

### Official Review · Reviewer_NH6L · 2021-07-16

**Rating:** 8
**Confidence:** 4

**Summary:**

The authors consider the paper of providing concise and approximately precise explanation for a black box binary function f on a specific instance x, where the explanation is a certificate for f(x), namely a small subsets of coordinates of x which determines f(x). The idea is the exploit decision trees T  that well approximate f and the possibility to implici construct only path of T used on instance x, using the tests on this path as a certificate.


**Limitations And Societal Impact:**

There are no potential negative social impacts to be considered.
Limitations (including weakness reported above) are discussed in the final section

**Main Review:**

The authors provides strong guarantees on the succintness and precision of an explanation that can be used to describe the behavior of a black box function f.
They exploit recent results on decision tree learning with universal guarantee (BLT,ICML20, BGLT,NeurIPS20] and Smyths's result on existence of decision tree approximating certificate complexity. The possibility of  implicitly building the decision tree with universal guarantee leads to the efficient construction of the local explanation. A strong point of the result is the fact that the underlying decision tree guaranteeing the approximate certificate complexity only depends on f and not on x.
Although the main technical contribution leading to main result are from previous papers, the overall contribution is strong enough to justify acceptance.

Strengths: the model inherits from previous works on which it leverage the leack of assumption on f.
Weakness: binary features and uniform distribution assumptions. The guarantee is with respect to the average certificate size. What could be said with respect to guarantee relative to the smallest certificate for the input instance x?

**Time Spent Reviewing:**

8

---

> ### Author Response · Authors · 2021-08-10
> **Response to Reviewer NH6L**
>
> Thank you for your review. We agree with your summary of our approach and our contributions.
>
> >binary features and uniform distribution assumptions
>
> Indeed our provable guarantees require feature and distributional assumptions; however, a core aspect of our framework is completely general: the reduction from certificate-finding to implicitly learning decision trees (all of section 2; in particular Lemma 2.1). We also note that there is a distribution-free algorithm for (non-implicit) learning of decision trees that runs in quasipolynomial time (Ehrenfeucht and Haussler, Learning decision trees from random examples, Information and Computing 89); however, it is not known to support the implicit-learning operations as efficiently as we require in this work. Finding implicit-learning guarantees for distribution-free algorithms, based on [EH89] or otherwise, is an exciting direction for followup work.
>
> >The guarantee is with respect to the average certificate size. What could be said with respect to guarantee relative to the smallest certificate for the input instance $x$?
>
> That’s a great question. As it turns out, approximating the size of the smallest certificate for a particular input is known to be $\mathrm{NP}^{\mathrm{PP}}$-complete [WMHK19 from our paper]. See line 76 for a discussion of this lower bound.

---

### Official Review · Reviewer_ikim · 2021-07-17

**Rating:** 8
**Confidence:** 4

**Summary:**

Given a definition of a succinct and precise certificate (or
explanation), the paper proposes an efficient algorithm for the
computation of such a certificate.

**Ethical Concerns:**

None.

**Limitations And Societal Impact:**

None.

**Main Review:**

The paper proposes an algorithm for computing succinct and precise
certificates (or explanations), which strong theoretical guarantees.
The proposed contribution is significant.

Concretely, the paper proposes to exploit recently proposed algorithms
for implicitly learning a decision tree for the generation of succinct
and precise certificates. The algorithm proposed in the paper is shown
in Figure 1, and builds on earlier work on implicit tree learning.


Additional comments:

The paper should include experiments supporting the proposed
algorithm.

Definition 1 seems equivalent to Definition 2.1 in:
S. Waldchen, J. MacDonald, S. Hauch, G. Kutyniok: The Computational
Complexity of Understanding Binary Classifier Decisions. J. Artif.
Intell. Res. 70: 351-387 (2021)
The paper cites an older version of this paper. It would be important
to cite the published paper, and of course clarifying whether the
definitions are the same.

The paper relates with [RSG18] (end of section 4.1). It shows that for
a concrete example, the proposed algorithm computes a better
certificate than [RSG18]. However, given other examples of the
limitations of [RSG18], could these be problematic for the work
proposed in this paper? One example is the assessment in:

A. Ignatiev: Towards Trustable Explainable AI. IJCAI 2020: 5154-5158

but also, in the papers cited by this paper.

Moreover, there is recent work related with explanations for decision
trees. The point is that decision trees may contain paths that are
arbitrarily larger than a succinct explanation. Could this in practice
impact the results proposed in the paper.

A few references on explaining decision trees:

Y. Izza, A. Ignatiev, J. Marques-Silva: On Explaining Decision
Trees. CoRR abs/2010.11034 (2020)

P. Barcelo, M. Monet, J. Perez, B. Subercaseaux: Model
Interpretability through the lens of Computational Complexity. NeurIPS
2020

G. Audemard, S. Bellart, L. Bounia, F. Koriche, J.-M. Lagniez,
P. Marquis: On the Computational Intelligibility of Boolean
Classifiers. CoRR abs/2104.06172 (2021)

X. Huang, Y. Izza, A. Ignatiev, J. Marques-Silva: On Efficiently
Explaining Graph-Based Classifiers. CoRR abs/2106.01350 (2021)

The paper analyzes the use of prime implicants as explanations and
argues that smallest prime implicants can be arbitrarily smaller than
plain prime implicants. This is correct. However, existing
experimental data suggests that the difference is not significant in
practice. For example, this is the case for neural networks:

A. Ignatiev, N. Narodytska, J. Marques-Silva: Abduction-Based
Explanations for Machine Learning Models. AAAI 2019: 1511-1519

**Time Spent Reviewing:**

3h

---

> ### Author Response · Authors · 2021-08-10
> **Response to Reviewer ikim**
>
> >The paper proposes an algorithm for computing succinct and precise certificates (or explanations), with strong theoretical guarantees. The proposed contribution is significant.
>
> Thank you for these kind words!
>
> >The paper should include experiments supporting the proposed algorithm.
>
> The focus of our work is to design a polynomial-time algorithm whose certificates are provably succinct and precise. We agree that an in-depth experimental comparison of our algorithm to alternatives is a concrete direction for followup work.
>
> >The paper cites an older version of this paper. It would be important to cite the published paper, and of course clarifying whether the definitions are the same.
>
> Thank you for pointing us to the published version -- we will update our citation.
>
> >One example is the assessment in… a few references on explaining decision trees:
>
> Thank you very much for the references. Indeed many of them are relevant and we will include a discussion of them in the next version of the paper.

---

### Decision · Program_Chairs · 2021-09-28

**Decision:**

Accept (Poster)

**Comment:**

Let me start by saying that throughout the discussion the reviewers all agreed that the paper discusses an interesting problem and makes a step towards creating succinct explanations. However, despite the fact that this work seems promising, it was judged as not quite ready for publication at this time.

We believe that if you revise the paper according to the comments provided by the reviewers then this work will undoubtedly evolve into a strong publication in the future.

Thanks for submitting your work to NeurIPS!

**Consistency Experiment:**

NeurIPS has a long history of experimentation. In 2014, NeurIPS ran an experiment in which 10% of submissions were reviewed by two independent committees to quantify the randomness in the review process. This year, we repeated a variant of this experiment to see how the quality of the review process has changed over time.  This paper was part of the experiment and was therefore assigned to two committees (consisting of reviewers, an Area Chair, and a Senior Area Chair) that reached independent decisions.  If both committees made the same recommendation, this recommendation was followed. If a single committee recommended acceptance, the paper was accepted (with the exception of a few cases in which the other committee identified what we considered a fatal flaw, e.g., an error in a key result).

This copy’s committee reached the following decision: **Reject**

The other committee assigned to the paper recommended **Accept (Poster)**.  You can find the other set of reviews, along with any follow up discussion with the authors here:
https://openreview.net/forum?id=9UjRw5bqURS